# Latitudinal Trend Analysis of Land Surface Temperature to Identify Urban Heat Pockets in Global Coastal Megacities

Dyutisree Halder [1,*], Rahul Dev Garg [1] and Alexander Fedotov [2]

1   Department of Civil Engineering, Indian Institute of Technology Roorkee, Roorkee 247667, India
2   Civil Engineering Institute, Peter the Great St. Petersburg Polytechnic University, 195251 St. Petersburg, Russia
*   Correspondence: dhalder@ce.iitr.ac.in

**Abstract:** Recent global warming has led to increased coastal disturbances through a significant transfer of heat between the land and the ocean surface. The polar regions show excessive temperature changes resulting in massive ice sheet melting. Mid-latitudinal storms pull heat away from the equator towards the poles; therefore, the global sea level is rising, making coastal cities the most vulnerable. In last few decades, rapid urbanization in big cities has drastically changed the land cover and land use due to deforestation, which has led to increased land surface temperatures (LSTs). This eventually leads to urban flooding due to oceanic storm surges frequently created by low pressure over the ocean during summer. This paper considered factors such as drastic unplanned urbanization to analyze coastal cities as the focal point of the generation of heat yielding the annihilation of the natural topography. Urban heat pockets (UHP) were studied for nine megacities, which were selected at an interval of 5° of latitudinal difference in the northern hemisphere (NH) since 70% of densely populated megacities are located in coastal regions. A comparative surface temperature analysis was effectively carried out with the same latitudinal reference for nine mid-sized cities using the derived LST data from Landsat 8. The results provide a comparative classification of surface temperature variations across the coastal cities over the NH. This study infers that the issues pertaining to growing urbanization are very important for analyzing the proportional impact caused by the settlement hierarchy and lays a robust foundation for advanced studies of global warming in coastal urban environments.

**Keywords:** global warming; urbanization; coastal megacity; Landsat 8; land surface temperature (LST); urban heat pocket (UHP)

## 1. Introduction

The urban population has been continuously migrating from rural areas to nearby cities since the beginning of the 19th century due to the industrial revolution and advancements in medicine, science, and engineering. In 2009, this transition created a new record when the urban population overtook the rural population for the first time in human civilization [1,2]. Currently, around 55% of the world's population lives in cities, and it is estimated that this percentage will rise to 60% by 2030 and to 68% by 2050 [3].

Global cities are observing a difference of opinion among conflicting urban future narratives. While many governments and business alliances have welcomed the idea that cities are the places where global capital has accumulated over the past three decades, forecasts of an urban ecological disaster have been found to increase through research in the fields of climate, water, and infrastructure [4]. Due to their high susceptibility to extreme weather events and poor ability to maintain flood management measures, coastal megacities in developing countries typically suffer the most [5].

The C40 Cities Climate Leadership Group suggested that coastal cities are most affected by the detrimental effects of climate change because roughly 75% of them are situated in coastal regions. They are immediately exposed to the risk of any future rise in sea level [6–9].

According to the standard choice of extreme coastal water level (ECWL) exposure analysis using SRTM and CoastalDEM, Climate Central, Princeton, USA, has broadly estimated in the recent literature that the global mean sea level is likely to rise 20 to 30 cm by 2050, and the recent projections incorporating Antarctic ice sheet dynamics indicate sea level rise of 100–180 cm by 2100 under representative concentration pathway (RCP) 8.5, which could even exceed 2 m or more in worst-case scenarios, but the chance of the sea level rising above 2 m is only 5%, according to experts' opinions. Sea level rise will continue with a 2.5-fold acceleration [10–13] if greenhouse gas concentrations continue their current patterns, which might have a significant impact on 300 million people [14]. This will provide favorable conditions for more destructive storms and rising rainfall rates as the average temperature rises along with the average sea level [15,16].

The latitudinal references with Earth rotation play a very important role in the changing seasons as well as in the assessment of climatic and weather disturbances all over the urban environment. In this study, megacities and mid-sized cities were taken into consideration with respect to these different latitudinal references to understand the variability in urban heating issues. Using a five-degree (5°N) interval in the northern hemisphere to study, classify, and identify urban heat pockets in coastal megacities gave a broader picture of the issues related to urban heating and its impact on climate change.

### 1.1. Why Megacities?

According to the United Nations, urban agglomerations with populations greater than ten million are classified as "megacities". Temperatures are generally higher than the surroundings in such areas because of the abundance of impervious surfaces that trap heat flux, where per capita vegetation is almost none [17]. According to the UN in 2019, 70% of the Earth's population will be urbanized by 2050, and the Earth's climate will continue to change. At the same time, the projected global warming and aggravated hydro-climatic extremes will hit megacities. As a result, the health and well-being of human populations and urban ecosystems will be at stake. These densely inhabited areas have accelerated global warming by producing significant amounts of radiation and reflection, which have an adverse effect on the local climate zone (LCZ). The local climate zone (LCZ) scheme is a good example of information-rich intra-urban classes that describe different types of urban land covers and land uses [18,19]. Climate-relevant urban data need to support risk assessment, and the right scale is an essential prerequisite for developing fit-for-purpose urban planning policies. The rapid urban demographic explosion is becoming a significant contributor in transforming the scales in cities. The rate of growth in regional and mid-sized cities has increased drastically since 1990; it will continue to grow and will probably become the next influential factor in development [20]. It is anticipated that over a third of the projected urban growth from now until 2050 will happen in three countries: India, China, and Nigeria. By 2050, it is projected that India, China, and Nigeria could add 416 million, 255 million, and 189 million urban dwellers, respectively [3].

### 1.2. Why Coastal Megacities?

Since the dawn of industrialization, it was observed that cities with seaports had the most benefits and vulnerability due to foreign elements. They were hubs for transferring goods as well as knowledge from other parts of the world but were also prone to adverse impacts from changing climatic behaviors [21]. These coastal cities have seen rapid growth due to their geographical locations. In recent years, they have also been hit by frequent cyclones, typhoons, and hurricanes all over the world [22]. It was previously predicted that changes in the frequency and intensity of storms have been brought by global warming because of increasing sea surface temperatures. Along with the wind damage that coastal storms generate, the flooding of low-lying areas and wave penetration into inland areas also have negative and devastating impacts [23–27].

Megacities such as Seoul, Kolkata, and Dhaka, which are not as close to the sea as other coastal megacities, are still affected by the same sea level rise and extreme weather

occurrences and are considered to be coastal cities. Sao Paulo is 800 m above sea level; therefore, even though it is less than 50 km from the shore, it is not a coastal city because of its elevation above sea level [28]. At present, out of 189 cities, the majority of which are coastal, including megacities such as Manila, Osaka, and Tokyo, a large number have a high risk of being affected by three or more different types of disasters. The number of coastal megacities has almost doubled in the last two decades. The total number of coastal megacities was 13 (out of 14 megacities) in 1990, which increased to 24 (out of 34 megacities) in 2018. The number is projected to be more than 50 by 2050 [3].

*1.3. Global Warming and Urbanization*

Global warming is making cities warmer, while urbanization accelerates the process via urban heat island (UHI) generation and aerosol radiative forcing [29]. By the middle of the 21st century, the consequences of interactions among climate change, temperature, air pollution, and the UHI effect are anticipated to increase the risk of poor human well-being and the mortality rate in cities globally [30]. Since 1950, the urbanized area has doubled in developed nations, whereas it has quintupled in developing nations [31]. The extra warmth of the built-up environment of a city is strictly connected to (i) the high heat storage capacities of building surface materials [32], i.e., asphalt and concrete, which reduce evapotranspiration and trap heat during the day and increase the heat storage capacity; (ii) urban morphology, e.g., narrow streets reduce air flow and automatically reduce the natural cooling effect; and (iii) anthropogenic heat production [33]. Global sea level rise is anticipated to accelerate in the 21st century as a result of human-caused global warming. The adverse consequences of climate change in coastal megacities include (a) an increased risk of flooding [34] and impeded drainage; (b) the salinization of freshwater supplies [35]; (c) higher water tables, which may reduce the safety of foundations; and (d) beach erosion [36].

The process of heating an environment begins with the interaction of incident solar rays with the Earth's surface. This solar radiation is absorbed, reflected, and re-radiated based on the surface properties of the incident surface. The amount of energy that a certain surface receives over a specific period is known as insolation [37]. Overpopulation and an expanding built-up environment boost the radiation and ends up contributing more heat to the atmosphere as the percentage of vegetation has reduced exponentially. Once the surface is heated, it contributes to heating the adjacent troposphere through a conventional method of heat transfer using pollutants in the air, which is very much proportional to the population density of any city.

Earth's climate was mostly influenced by the sun and volcanic eruptions before the industrial revolution. Since recent massive anthropogenic interventions in developing the built environment all over the world, the number of heat waves has increased drastically. During monsoons, some places experience much heavier rainfall than they used to receive. Coastal megacities are becoming more vulnerable due to sea level rise, which leads to the loss of countless lives. This approach is well researched so far, but on the contrary, coastal megacities are the primary source of massive heat due to the use of non-sustainable built-up materials and greenhouse gas emissions through unplanned urbanization.

Therefore, the key aim of this study was to identify the behavior of existing land use in large megacities that perform differently than mid-sized cities. Using a latitudinal interval of 5 degrees to select megacities and mid-sized cities provided an overall global perspective to the problem of urban heating. The aim was attained through the following objectives: (i) quantifying the LST for nine megacities and nine mid-sized cities using thermal data acquired from Landsat 8, (ii) binary classification of remotely sensed images using support vector machine (SVM) classification for all 18 cities, (iii) assessing the correlation between LST and the SVM-based classified outputs, and finally, (iv) mapping and quantifying urban heat pockets (UHP) based on the heat radiation level for each megacity with respect to the paired mid-sized city.

## 2. Data and Methodology

The United Nations has listed the megacities globally. According to their coastal locations, nine large coastal megacities were identified at an interval of almost 5° of latitudinal difference in the northern hemisphere to obtain a worldwide perception of the surface temperature analysis from the equator to the Arctic Circle. Mid-sized cities at similar latitudes were considered in this study for comparison, where all the selected cities and megacities were located at an elevation varying from 0 m to 41 m from the mean sea level and within 100 km of the shoreline (Figure 1). Population data were collected from censuses with respect to areas of different boundaries. This research only considered the core city boundaries, where urban agglomeration is extreme and most of the population is highly exposed to urban natural calamities [38].

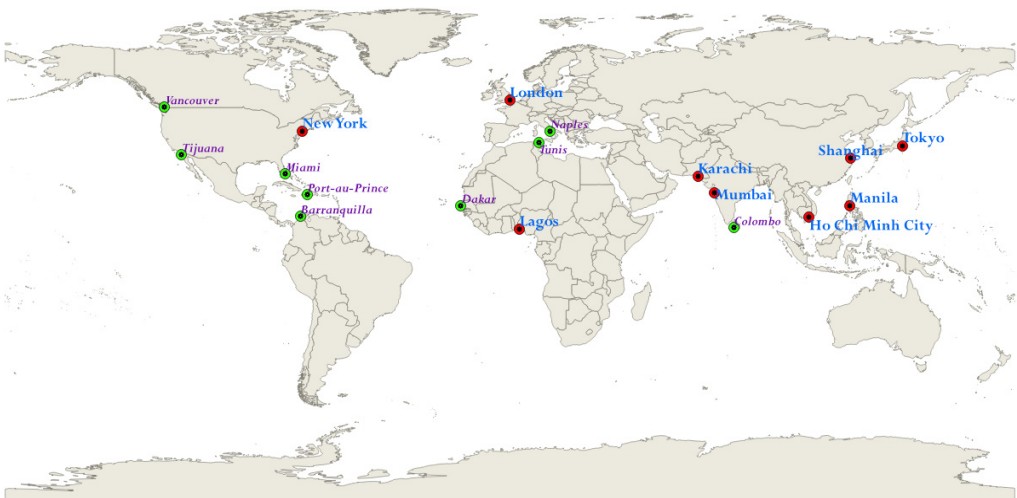

**Figure 1.** Coastal megacities and cities selected for the study (red dots: megacities, green dots: mid-sized cities).

For each coastal megacity and city, according to their hottest month of the year, the Landsat 8 OLI and TIR 30 m dataset was collected from the USGS website [39]. In this step, the downloaded dataset was radiometrically corrected and further used to extract the radiative skin temperature, commonly known as the land surface temperature (LST), of the selected date for each megacity and city. The LST extraction provided the result of the mentioned time (Table 1), which was performed through the translation of the DN (digital number) to $L_\lambda$ (spectral radiance) using the equations mentioned in Section 2.1 from the Landsat user's handbook. The detailed workflow of the methodology of this study is demonstrated in Figure 2.

Using the Landsat 8 data, atmospheric and radiometric corrections were carried out as a preprocessing task for the acquired data. The administrative city boundaries for both megacities and mid-sized cities were used in the form of shape files to clip the downloaded datasets. Then, the brightness temperature was extracted using thermal band 10, and the NDVI was generated from band 4 and band 5 to compute the LST for all the cities and megacities. This whole process was executed using the raster calculator function in QGIS software. Shape files for the boundaries of the megacities and cities were downloaded from the respective administrative portals. Afterwards, supervised classification was performed over all the datasets using an SVM algorithm, which performed better than all other existing classification algorithms, mainly in densely populated urban areas [40–42]. For each image, 100–300 region-of-interest (ROI) samples were taken according to the city sizes. Each ROI included more than 100 pixels for each class. The classes for the training samples were built and unbuilt. To perform this task, the ENVI was used. After this, the classified data were exported in QGIS for further analysis.

**Table 1.** List of cities and megacities with their latitudes/longitudes, elevations, populations, areas, and Landsat 8 date and time acquisition data.

| Megacities/Cities | Latitude/Longitude | Elevation | Population | Area (km$^2$) | Date Acquired | Time Acquired |
|---|---|---|---|---|---|---|
| Lagos Metropolis, Nigeria | 6°27′N/3°24′E | 41 | 13,432,000 | 999 | 21/2/2020 | 11:02:56 |
| *Colombo, Sri Lanka* | 6°55′N/79°59′E | 1 | 752,993 | 37 | 11/4/2020 | 10:23:33 |
| *Barranquilla, Columbia* | 10°57′N/74°47′W | 18 | 1,212,943 | 154 | 30/4/2020 | 10:16:24 |
| Hồ Chí Minh (HCM), Vietnam | 10°45′N/106°39′E | 19 | 7,004,921 | 494 | 13/3/2021 | 10:13:43 |
| Metro Manila (NCR), the Philippines | 14°35′N/120°59′E | 3 | 13,484,462 | 619 | 17/5/2021 | 10:17:01 |
| *Dakar, Senegal* | 14°43′N/17°28′W | 22 | 1,146,052 | 83 | 27/10/2020 | 11:33:58 |
| *Port-au-Prince, Haiti* | 18°31′N/72°17′W | 15 | 987,310 | 36 | 20/8/2020 | 10:15:00 |
| Greater Mumbai, India | 19°04′N/72°52′E | 14 | 12,478,447 | 458 | 28/5/2019 | 10:33:24 |
| Karachi (KSDP), Pakistan | 24°51′N/66°59′E | 10 | 15,400,253 | 1890 | 26/5/2020 | 10:56:10 |
| *Miami, Florida* | 25°45′N/80°8′W | 2 | 467,963 | 143 | 9/8/2018 | 10:49:24 |
| Shanghai, China | 31°13′N/121°28′E | 4 | 25,582,895 | 6833 | 16/8/2020 | 10:25:01 |
| *Tijuana, Mexico* | 32°31′N/117°2′W | 20 | 1,810,645 | 291 | 29/8/2020 | 10:22:41 |
| Greater Tokyo, Japan | 35°39′N/139°50′E | 40 | 9,300,421 | 628 | 6/8/2021 | 10:15:55 |
| *Tunis, Tunisia* | 36°48′N/10°10′E | 4 | 638,845 | 212 | 2/8/2021 | 10:54:38 |
| Greater New York, US | 40°43′N/73°56′W | 10 | 8,804,190 | 784 | 6/7/2020 | 10:39:41 |
| *Napoli (Naples), Italy* | 40°51′N/14°18′E | 17 | 967,068 | 119 | 7/7/2020 | 10:47:25 |
| *Vancouver, Canada* | 49°15′N/123°6′W | 2 | 631,486 | 114 | 14/8/2020 | 11:01:26 |
| Greater London, UK | 51°30′N/0°7′W | 11 | 9,002,488 | 1572 | 15/7/2018 | 10:51:33 |

All the built-up pixels, in form of a point feature (raster to vector), for each classified dataset were extracted in QGIS using the raster to vector function. The purpose of this task was to separate heat generated from a built-up environment from any other classes such as vegetation, waterbodies, and barren land (rock, soil, and sand) because in some places barren land shows higher LST values than built-up environments, and our main concern was to study the LST caused by built-up environments, which mainly consist of buildings, roads, and other manmade features. For all those built-up point features, the LST was added as an attribute from the initially derived LST images for each city and megacity by sampling those points from the LST raster data. After attaching those LST DN values to the point features, they were reclassified into five classes based on the temperature, which were defined as <20, 20–25, 25–30, 30–35, and >35. Finally, this thematic reclassification was overlaid on the SVM-classified images of all respective cities and megacities to generate the final maps of urban heat. The surface temperatures for different materials throughout the day were taken for 3 consecutive days in the last week of May 2022 and were acquired to validate findings in the Mumbai area.

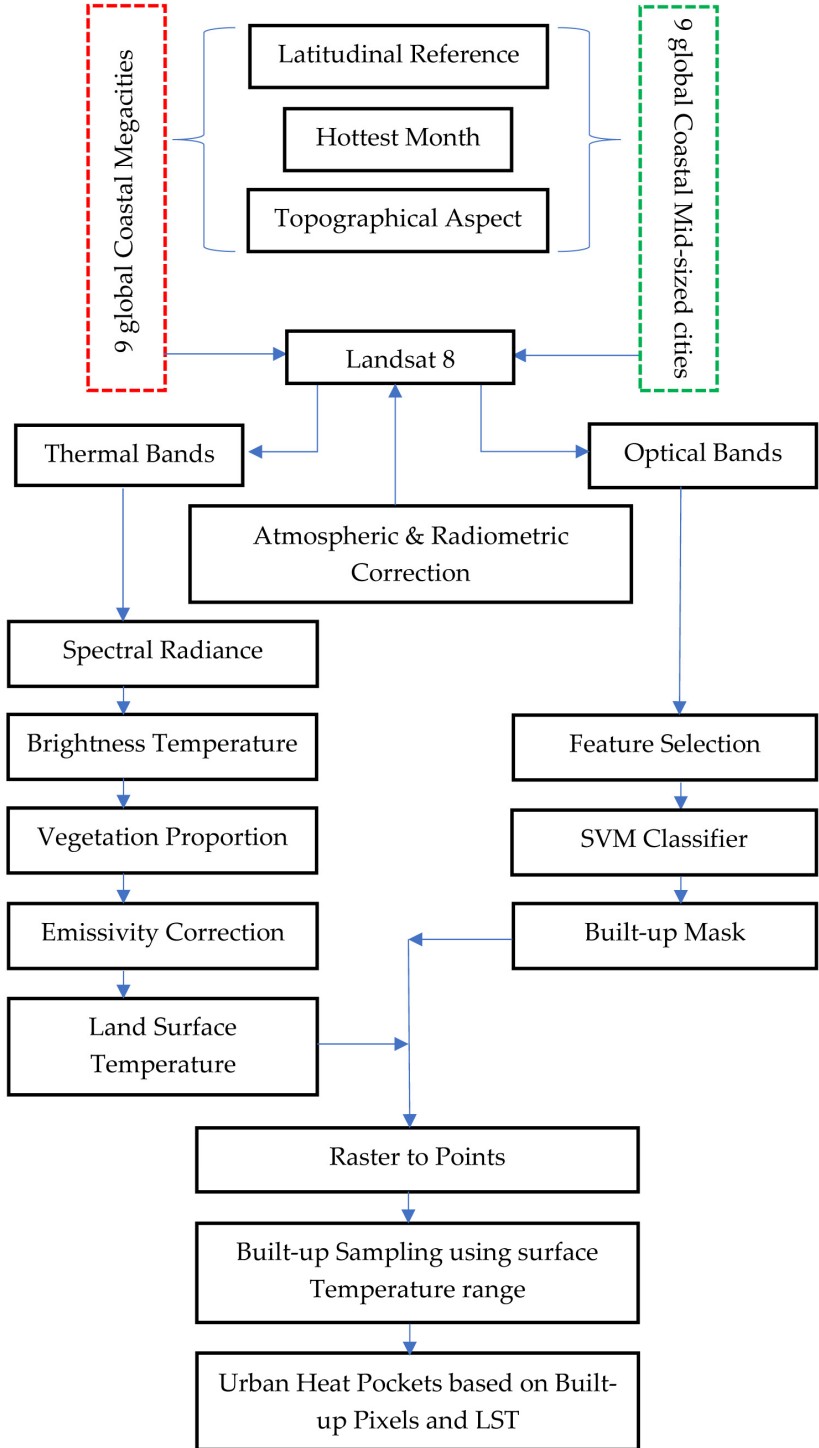

**Figure 2.** Methodological flowchart of the study.

## 2.1. Land Surface Temperature Extraction

The satellite sensors receive electromagnetic thermal energy using their thermal sensors and store it as digital numbers (DN). This analysis was performed in QGIS software for each image.

$$L_\lambda = ML * Band\_10 + AL - O_i \tag{1}$$

where $L_\lambda$ = at-sensor spectral radiance; ML represents the radiance multiplier of band No. 10; AL = radiance of band No. 10; and Oi represents the correction value for band 10, which is 0.29. For OLI-TIRS (Landsat 8), the ML for band 10 is 0.0003342 and the AL is 0.10000.

Spectral radiance to brightness temperature conversion in Kelvin (K):

$$BT = [K_2 / \ln(K_1 / L_\lambda + 1)] - 273.15 \qquad (2)$$

BT represents the satellite brightness temperature in Kelvin; $K_1$ represents calibration constant 1; and $K_2$ represents calibration constant 2. The calibration constants for OLI-TIRS (Landsat 8) $K_1$ and $K_2$ are, respectively, 774.8853 and 1321.0789. Then, BT is converted to Celsius.

Emissivity of surface temperature correction:

Emissivity is defined as difference between the electromagnetic radiance of an object and a blackbody. It is not only dependent upon the surface materials (i.e., the surface's physical and chemical properties) but also upon the surface roughness of the object.

$$NDVI = (NIR - Red) / (NIR + Red) \qquad (3)$$

In this study, NDVI (normalized difference vegetation index)-based emissivity measurement for Landsat 8 was considered, where band 5 was considered as a near-infrared band (NIR) and band 4 was considered red.

$$PV = [(NDVI - NDVI_{Min}) / (NDVI_{Max} - NDVI_{Min})]^2 \qquad (4)$$

where PV is the proportion of vegetation extracted from Equation (4). For each city, the mean, minimum, and maximum NDVI values were used for PV extraction.

$$E = m * PV + n \qquad (5)$$

where m and n are, respectively, 0.004 and 0.986.

Brightness temperatures can lead to errors in the computation of surface temperature. In order to minimize these errors, emissivity correction is important and is performed to retrieve the LST from BT using Equation (6), where LST is in Celsius (C).

$$LST = BT / (1 + (\lambda * BT / \alpha) * \ln(\varepsilon)) \qquad (6)$$

where $\lambda$ represents the emitted radiance wavelength for band 10, i.e., 10.8 μm; $\alpha$ represents $1.438 \times 10^{-2}$ mk; and $\varepsilon$ represents the land surface emissivity ($\alpha$ is calculated as $\alpha = hc/\sigma$, where h represents Plank's constant ($6.626 \times 10^{-34}$ Js), c represents the velocity of light ($3 \times 108$ m/s), and $\sigma$ represents the Stefan–Boltzmann constant ($1.38 \times 10^{-23}$ J/K)) [43,44].

### 2.2. SVM Classification for Urban Built-Up Environment Extraction

The most prevalent method for classifying satellite images is pixel-based classification, which uses quantitative approaches to discern patterns per pixel inside an image. Apart from pixel-based classification, another type is called object-based classification, in which rather than taking pixels as the minimum unit, it divides the image into objects and uses the spectral, spatial, contextual, and textual characteristics between them to classify them [45–47]. Pixel-based classification, a traditional classification that uses the combined spectral responses of all pixels in a training set for a given class, is considered very effective for low to moderate spatial resolution data [48]. For mapping a complex urban environment in a city, the SVM classifier achieved higher overall accuracy than the maximum likelihood classifier across all other classification schemes [49,50]. SVM is a supervised classification method derived from statistical learning theory that generates better classification outputs from complicated and noisy data. With a decision surface that optimizes the margin between the classes, it conceptually divides the classes. An SVM-based classification using kernel-type variables such as RBF (radial basis function) was performed in ENVI software to create a binary image with built and unbuilt classes.

### 2.3. Assessment of Urban Heat Pockets

The SVM-classified raster was used to extract the LST values for built-up pixels by overlaying the results with the computed LST data. The frequency of built-up pixels in the five classes of temperature range was computed to extract the heat variations in built-up environments. Based on these heated zones, urban heat pockets (UHP) were quantified (Table 2).

$$UHP_i\% = f_i/n \tag{7}$$

where f is the frequency of built-up pixels for i = I, II, III, IV, and V, which depict classes based on computed LST, and n is total No. of built-up pixels. UHP classes I, II, III, IV, and V varied, respectively, from LST pixel values, ranging from >35 °C to 35 °C–30 °C, 30 °C–25 °C, 25 °C–20 °C, and <20 °C.

**Table 2.** Classification of urban heat pockets.

| UHP Class | LST Pixel Value | Severity |
|---|---|---|
| I | >35 °C | Extreme heat stress |
| II | 35 °C–30 °C | Moderate heat stress |
| III | 30 °C–25 °C | Slight heat stress |
| IV | 25 °C–20 °C | Comfortable heat stress |
| V | <20 °C | No thermal stress |

## 3. Results

The LST maps for each coastal megacity and corresponding mid-sized city of similar latitudinal reference are shown in Figures 3–5, with examples from lower-, mid-, and upper latitude regions. Different shades of red show the temperature, ranging from <20 °C to >35 °C, as per the LST recorded for that particular mid-sized city and megacity in same latitudinal reference. All the other pairs of coastal megacities and corresponding mid-sized cities are attached in Appendix A (Figures A1 and A2).

Observations made from the histogram (Figure 6) were investigated and are presented in form of image pairs. A comparative analysis (Table 3) provides a better understanding of the variation in the latitudinal references and their heat signatures.

**Table 3.** Built-up area percentage, highest and mean LST, and elevation and azimuth sun angles for cities and megacities.

| Megacities/Cities | Built-Up Area (%) | Highest LST of Built-Up Area (°C) | Mean LST of Built-Up Area (°C) | Solar Elevation Angle | Solar Azimuth Angle |
|---|---|---|---|---|---|
| **Lagos** | 69.44 | **33.39** | **30.25** | 52.18 | 128.62 |
| *Colombo* | 75.41 | 31.56 | 27.86 | 67.95 | 84.52 |
| *Barranquilla* | 55.93 | 34.78 | 28.58 | 65.02 | 77.91 |
| **Hồ Chí Minh** | 61.88 | **35.02** | **28.96** | 59.95 | 115.47 |
| **Manila** | 61.96 | **41.26** | **35.21** | 66.25 | 75.01 |
| *Dakar* | 80.03 | 39.91 | 32.61 | 55.41 | 142.26 |
| *Port-au-Prince* | 53.44 | 34.66 | 30.11 | 51.61 | 93.76 |
| **Mumbai** | 40.16 | **41.36** | **33.23** | 63.26 | 80.08 |
| **Karachi** | 25.59 | **40.54** | **33.79** | 68.29 | 95.59 |
| *Miami* | 74.08 | 35.43 | 31.57 | 51.15 | 96.81 |
| **Shanghai** | 34.83 | **44.01** | **29.07** | 58.87 | 126.32 |
| *Tijuana* | 78.48 | 42.98 | 34.55 | 52.95 | 122.45 |
| **Tokyo** | 79.83 | **41.02** | **33.27** | 62.14 | 126.97 |
| *Tunis* | 58.19 | 40.40 | 32.35 | 63.11 | 129.76 |
| **New York** | 67.21 | **45.91** | **33.39** | 56.27 | 111.92 |
| *Naples* | 70.83 | 38.90 | 31.71 | 55.87 | 110.43 |
| *Vancouver* | 76.56 | 33.00 | 24.85 | 45.04 | 129.45 |
| **London** | 68.29 | **45.90** | **32.72** | 49.45 | 124.27 |

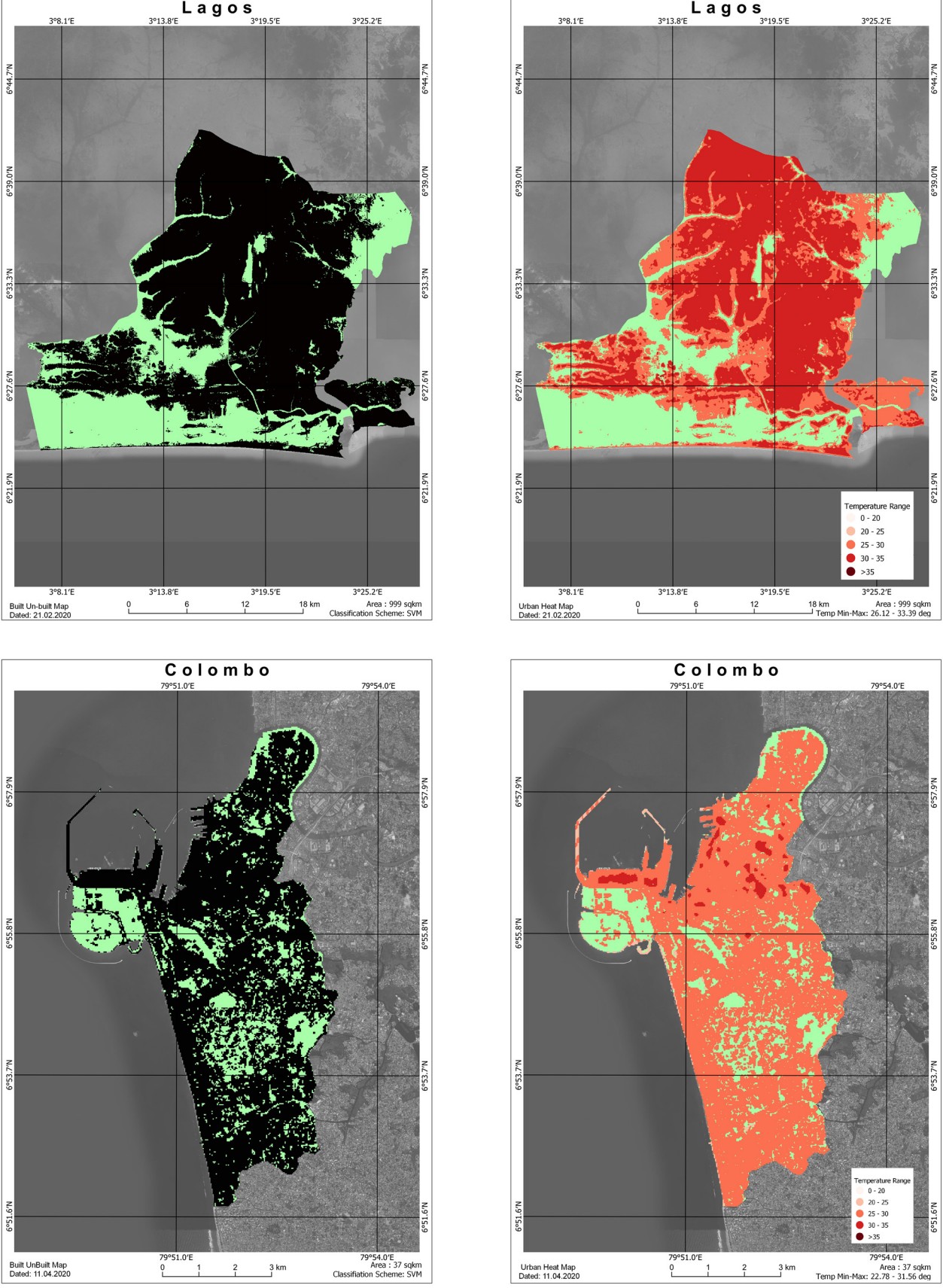

**Figure 3.** Built (black)/unbuilt (green) map with a corresponding heat map for both a megacity and its respective mid-sized city in a low-latitude region (Lagos and Colombo).

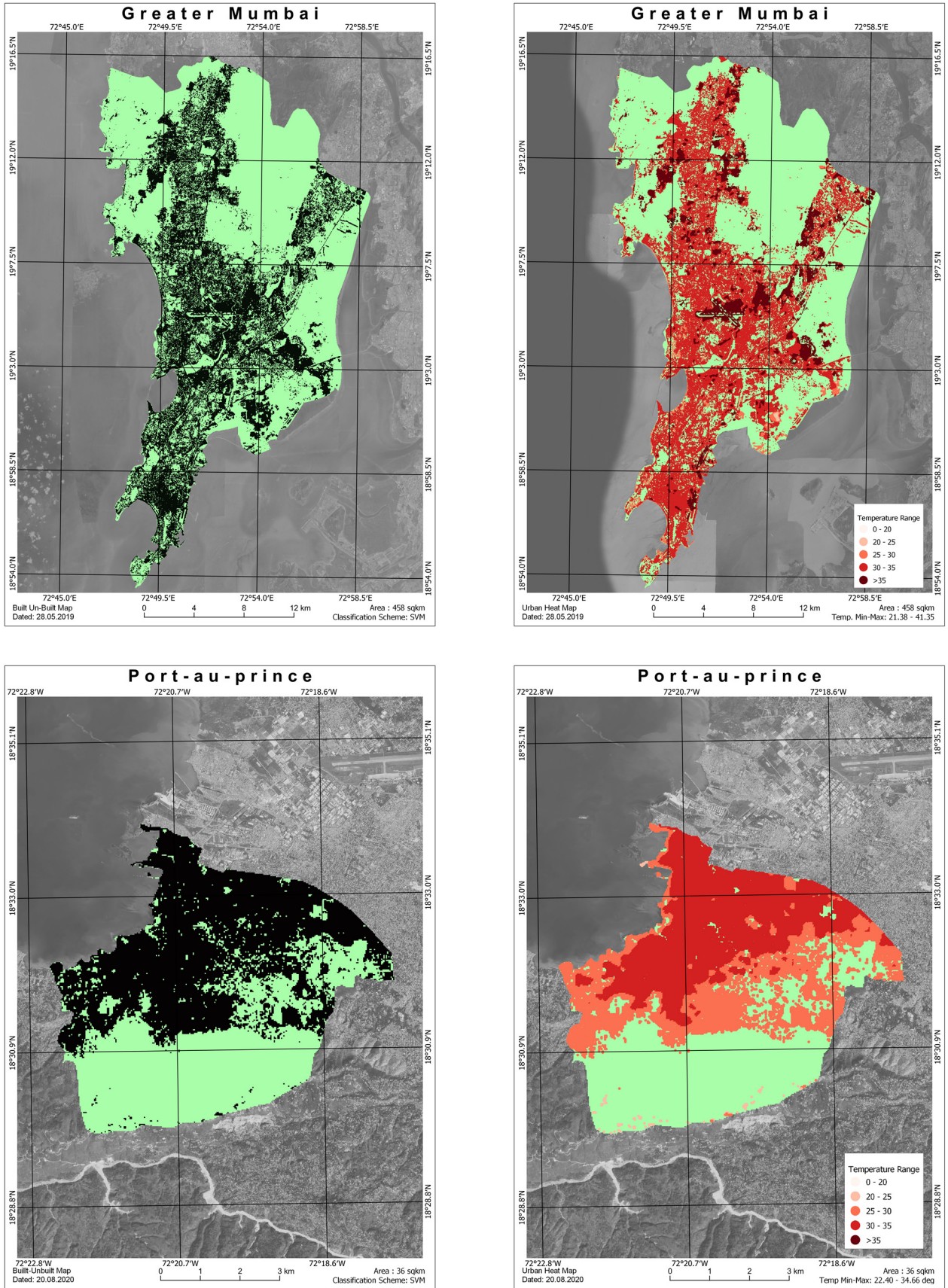

**Figure 4.** Built (black)/unbuilt (green) map with a corresponding heat map for both a megacity and its respective mid-sized city in mid-latitude region (Mumbai and Port-au-Prince).

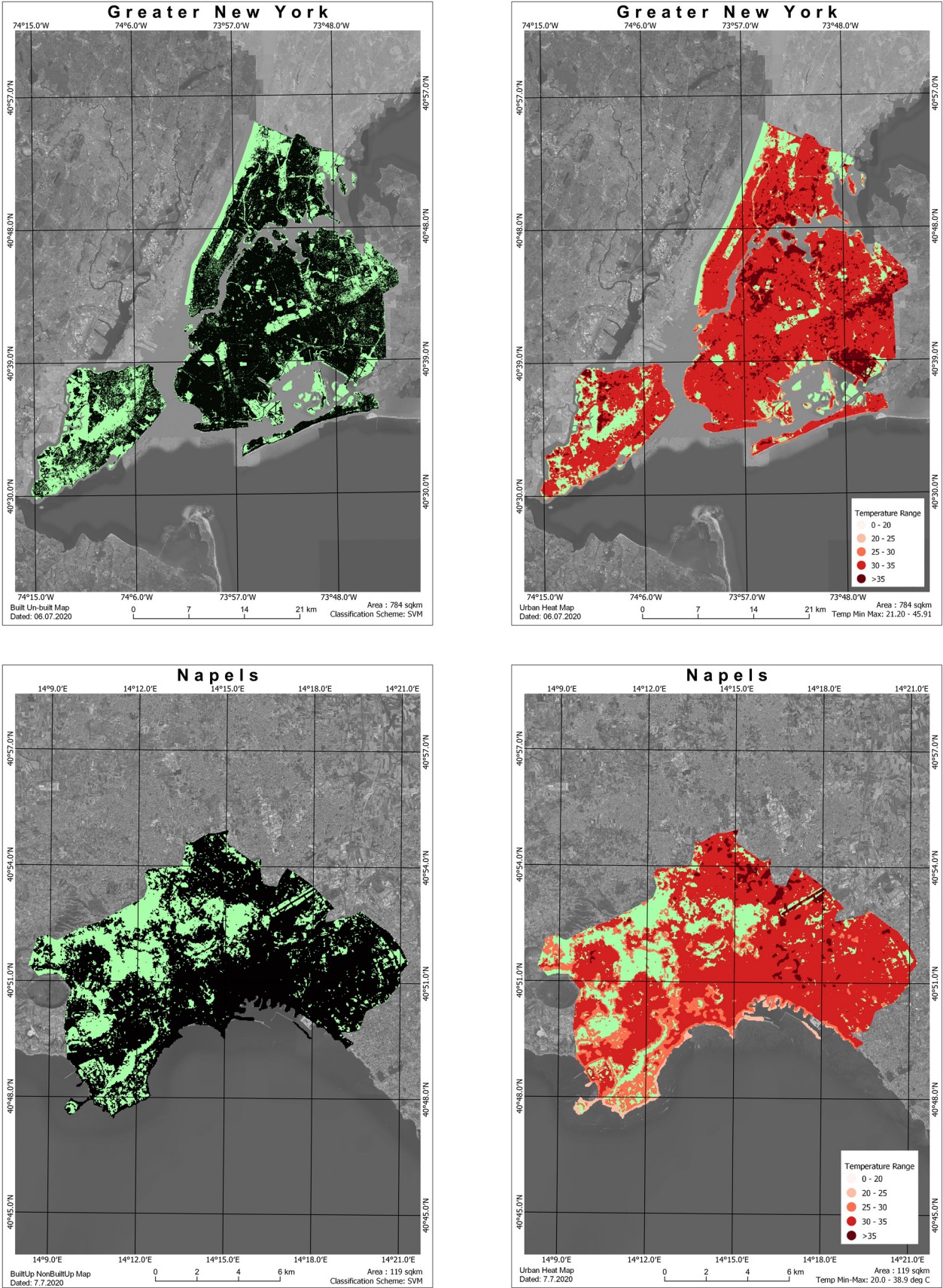

**Figure 5.** Built (black)/unbuilt (green) map with a corresponding heat map for both a megacity and its respective mid-sized city in a high-latitude region (New York and Naples).

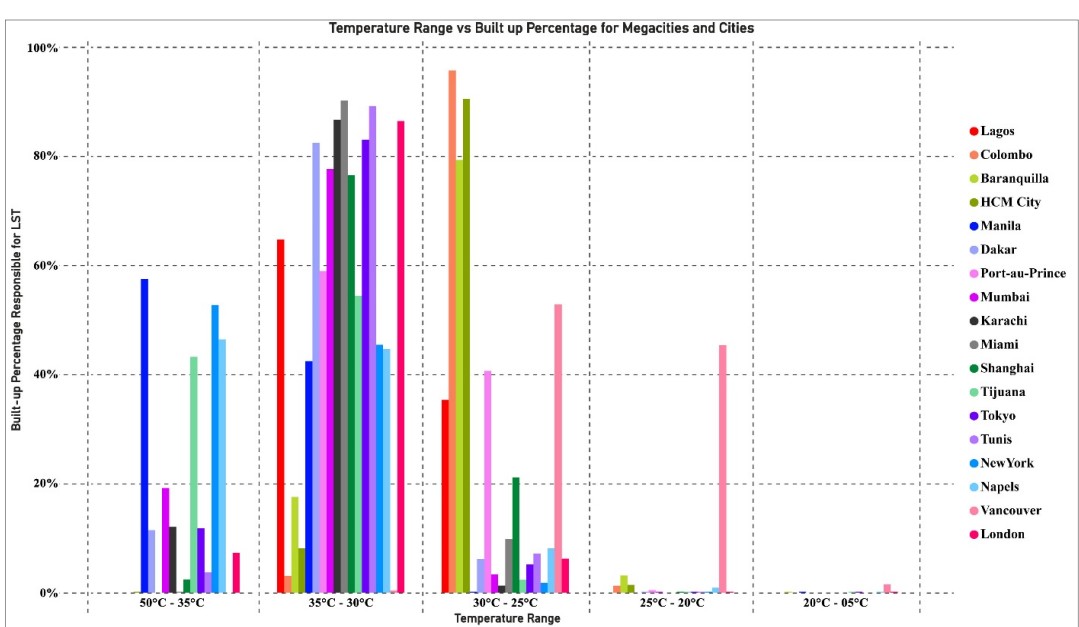

**Figure 6.** Temperature range and built-up percentage for each city and megacity.

*Lagos and Colombo*—The megacity Lagos was observed to have a highest LST 2 °C higher than Colombo. Both are capital cities, and they share nearly same latitude. The mean LST was 2.5 °C higher in Lagos, whereas the sun elevation angle is somewhat lower compared to Colombo in their respective summers. However, on a brighter side, Lagos did not have any built-up area showing an LST >35 °C.

*Barranquilla and HCM city*—The megacity HCM and Barranquilla, the capital city of Colombia, were observed to have similar thermal performances. The LST difference was negligible, and being a megacity, HCM was only higher by 0.24 °C in terms of the highest LST and by 0.38 °C in terms of the mean LST.

*Manila and Dakar*—Manila is one of the most densely populated megacities in the world. It highest LST was almost 1.5 °C higher than that of Dakar, which is of similar density. Moreover, the mean LST of Manila was 2.5 °C higher than that of Dakar. Additionally, 57.46% of the total built-up environment had an LST higher than 35 °C.

*Port-au-Prince and Mumbai*—The highest LST of Mumbai was much higher than that of Port-au-Prince, and the city was loaded, with 19.17% of the total built-up area having an LST of more than 35 °C, but the mean LST difference was smaller in a brighter context.

*Karachi and Miami*—Karachi also had a difference of around 5 °C for its highest LST compared to Miami, and 12.04% of the total built-up area was higher than 35 °C.

*Shanghai and Tijuana*—Shanghai was observed to have a highest LST only 1 °C higher than that of Tijuana. However, surprisingly, Tijuana was noticed to have a mean LST of 34.5 °C, which was 5.5 °C higher than that of Shanghai. Additionally, in Shanghai, only 2.40% of the total built-up area was higher than 35 °C, whereas this number for Tijuana was 43%, which signifies the alarming level of heat generation inside the city.

*Tokyo and Tunis*—Tokyo is one of the megacities where the population is decreasing, but the city is already densely populated, and Tunis is the capital city of Tunisia. Tokyo was observed to have a higher mean LST than Tunis. Additionally, Tokyo had a higher percentage of its total built-up area higher than 35 °C.

*New York and Naples*—NYC was observed to have a difference of 7 °C in its highest LST compared to Naples. It was only 2 °C higher in terms of the mean LST, and both had almost 50% of their total built-up areas higher than 35 °C.

*Vancouver and London*—The highest LST in London was almost 13 °C higher than that of Vancouver, and the mean LST was 8 °C higher. Additionally, London had 7.25% of its total built-up area higher than 35 °C.

## 4. Discussion

High summer-time temperatures are arriving earlier and lasting longer, mostly in the northern hemisphere because the climatic zones are shifting towards the equator [51], and coastal megacities are extremely exposed to natural calamities due to this phenomenon [28,38,52]. Therefore, the first approach was to choose cities with larger percentages of urban population, which will naturally cause a high percentage of built-up areas (megacities were considered according to UN dataset). While studying the global megacities, it was observed that 70% of the total megacities are located on the coast since coastal areas have maximal access in terms of transportation. In particular, most of the cities developed since European colonization, which made it easy for further development in the post-industrialization era.

Megacities were selected globally, as the approach was to justify the global perspective. Therefore, nine coastal megacities were identified worldwide at different longitudes, from Tokyo at the extreme east to New York at the extreme west. Based on the solar flashlight effect theory, the selected cities were categorized based on their latitudes, as the sunlight angle is different from the equator to the poles [53]. At the peak of summer, at a specific time, sunlight will fall on the places at an angle of 90°; thus, the solar insolation will be maximal, and the land surface temperature will be at its highest from the equator to the tropics. However, from the tropics to the poles, this phenomenon never happens, even at the peak of summer because of inclination of the Earth's rotation has an angle of 23.5° and sunlight is received at an angle less than 90°. Based on this fact, at the same time, insolation will vary from the equator to the poles in megacities at different latitude on Earth, and the places with same latitude will have the same insolation. Thus, the approach was to pair nine megacities of the northern hemisphere with nine mid-sized cities to observe and analyze whether the surface thermal performances of the cities were the same or different.

According to our methodology, the peak summer months were identified for those 18 megacities and cities. However, summertime was not same for the cities at the same latitudes because of local or regional climatic factors such as oceanic currents and the wind pressure direction. Landsat 8 data were downloaded for each megacity and city, which were selected earlier based on their hottest months. While downloading, it was difficult to obtain cloud-free datasets, as coastal cities have the problem of frequent cloud formation due to being near to the oceans. A summer 2020 timeline was considered for LST extraction. In some cases, images were taken from the summer of 2021/2019/2018 due to the non-availability of cloud-free data.

As per the research, the outdoor comfortable temperature is 26.2 °C, which is fundamentally an air temperature (AT), but the acceptable outdoor temperature with slight heat stress ranges between 26.2 °C and 31.6 °C [54]. However, the temperature does not affect the comfort level alone since the relative humidity (RH) and wind play a vital role in maintaining a comfortable outdoor environment, and a 26.2 °C outdoor air temperature at 40% humidity is the ideal condition [55–57]. Nevertheless, if humidity reaches beyond 70%, the same air temperature feels so much hotter because the body's sweat evaporates slowly when the air is already saturated with water. A metric called the heat index indicates that if the body experiences 88 °F (32 °F = 0 °C) weather with 85% humidity, then it feels like the temperature is 110 °F [58]. Air adjacent to the surface is heated by radiation and conduction, so longer day hours with an elevated sun angle on a particular area might increase insolation. Additionally the emissivity of outdoor materials and the outdoor environment reaches its intolerable temperature in the afternoon between 1 pm and 3 pm due to radiation and re-radiation through the pollutants present in the air [59,60]. In conclusion, higher emissivity means a higher LST, which means a higher outdoor air temperature.

On a hot summer day in the afternoon, the land surface temperature (LST) and the air temperature (AT) might vary by up to 9 °C [61]. In justification of the above statement, a field study of the surface temperature variation was conducted in one of the listed cities (Table 1). The megacity Mumbai experienced various stages of development for last three centuries, so the built-up environment includes an extensive variety of building materials

with different emissivity [62]. The surface temperatures for different buildings, bridges, roads, parks, etc., were recorded for three consecutive days in the last week (peak summer days) of May 2022 due to the higher UHI intensity measurements that were taken on the mentioned days [63]. The experiment was planned for the time of our observation to overlap with the data acquisition time of Landsat 8 of the study area, which was around 10:30 a.m. IST for the Mumbai region.

In Figure 7, it is observed that the average difference in LST at 10 a.m. and 2 p.m. was 10.4 °C, so for example, in an area in Mumbai, if the LST is 35 °C at 10 a.m., then it will be around 45.4 °C at 2 p.m., and as per the previous reference [61], the air temperature at same time should vary between 36.4 °C and 45.4 °C. The LST was significantly higher than the air temperature since the majority of the city's areas are built-up and impermeable, which traps heat from the sun [44]. As a coastal city, the RH of Mumbai will always vary between 55% and 80% throughout the day, as per the Power Lark data. As a result, it was decided to segregate the built-up areas in different urban heat pockets (UHP), with LST > 35 °C as class I, 35 °C–30 °C as class II, 30 °C–25 °C as class III, 25 °C–20 °C as class IV, and <20 °C as class V.

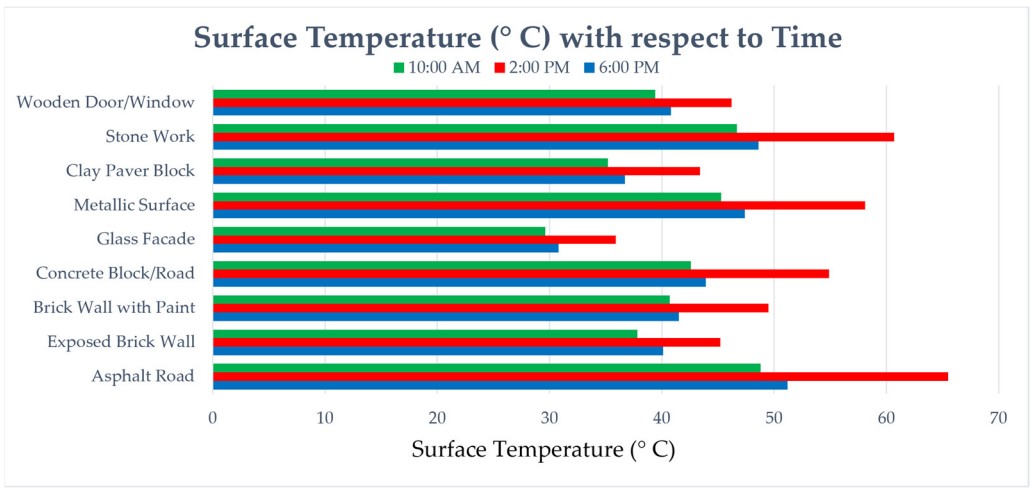

**Figure 7.** Recorded daily temporal data for the surface temperatures of various built-up materials in Mumbai.

In reference to Figure 8, the megacities Lagos and Ho Chi Minh showed some similarities with respect to their similar latitudinal counterparts. UHP Class I was not observed in Lagos, and with respect to other megacities, the outdoor temperature was bearable because in megacities such as Lagos, where the population density is quite high, people live without shelter. There are two components that might be responsible for this kind of behavior: a low-rise built-up area using locally available building materials, and mixed land use in urban areas. The former component validates Lagos's lower LST, while later suggests a case for HCM and its low-LST behavior.

Manila showed a much higher percentage in class I of UHP, which is quite alarming. As far as the reason for this behavior is concerned, in Manila, most of the built-up surface's materials, such as concrete and glass, are observed. This makes the city's heat balance quite vulnerable, while Dakar did not show much of a difference in its mean LST because of the influence of the surrounding deserted area.

Mumbai and Karachi had around 2 °C to 3 °C differences in their mean LSTs compared to their respective cities, i.e., Port-au-Prince and Miami. As per the graph shown in Figure 8, it can be concluded that both megacities had very low percentages of their built-up areas responsible for temperature ranges greater than 35 °C. This pattern could be seen because of the recent developments in the last decade in the built-up region because of rapid urbanization. Buildings are mostly made of materials such as glass and concrete and are of

the high-rise type, with artificial cooling, which increases greenhouse gas emissions and heats up the overall outdoor environment.

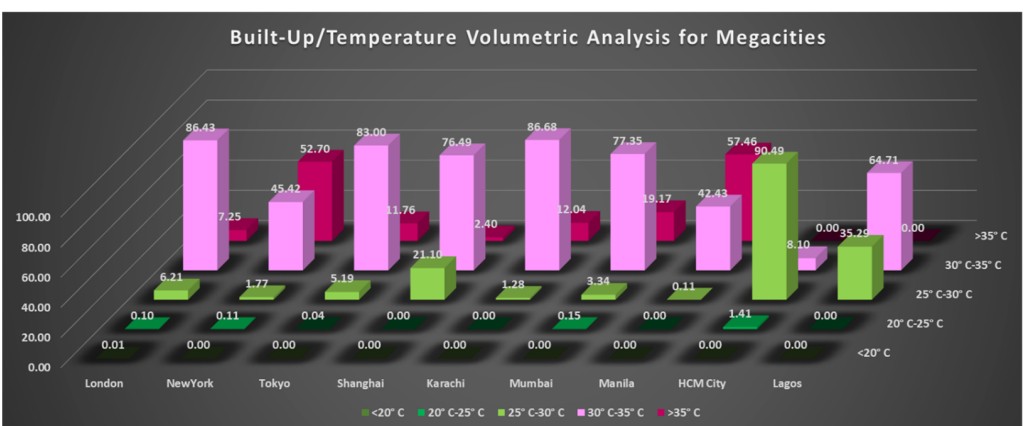

**Figure 8.** Visualization of volumetric analysis of UHP classes of megacities.

Shanghai is a well-planned megacity with a lot of mixed land use comprising both built-up areas (impervious surfaces) and vegetation in parallel. This pattern reduces the surface temperature pixel value (30 × 30 m) more than a pixel mostly with built-up environment [64]. While the topography in Tijuana has a higher percentage of barren hills, which reflect, the built-up area becomes extremely hot because of multiple sources of radiation. The observed difference in the mean LSTs of these two was much higher, and Tijuana showed a higher mean LST compared to other city–megacity pairs. This shows that the topographical properties of surroundings could be taken as one of the prominent factors for understanding the urban heat in a city. Like Shanghai, the well-planned Tokyo benefits from its topographical features (urban forest), which make it cooler than the other surrounding cities of neighboring countries. Natural surfaces, i.e., vegetation and water, that have high emissivity are one of the prominent mitigating factors in the case of Tokyo's urban heating. While Tunis is not a megacity, it still shows behavior similar to one due to its surrounding natural desert feature on one side, which heats up the city during the day. The topography can be seen to play a predominant role in increasing heat as well as mitigating urban heat.

In case of New York, being an international hub for business and cultural integration, the city has a lot of high-rise buildings, which are constructed of steel and concrete. This explains the high percentage of class I UHP. Whereas Naples shows the same behavioral pattern and reason for being, it mostly has old stone buildings, which reflect more heat in the daytime and thus have a significant impact on a high LST [65]. As London is the oldest megacity, growth-wise planning can be seen as well as different building materials. It was observed that newly constructed built-up areas in London showed behavior of UHP class I, whereas Vancouver can be called one of the coldest cities, even during summer.

## 5. Conclusions

In this paper, the impact of coastal megacities on global warming was analyzed, as two thirds of all megacities are situated on the coast. Based on the solar flashlight effect, nine megacities were selected at a latitude interval of 5° in the northern hemisphere and analyzed using a Landsat 8 dataset. The results were derived through LST extraction and SVM classification of these images, which showed the correlation between the built-up percentage and urban heat pockets for each megacity. In parallel, nine mid-sized cities were also taken under observation to understand the difference in heat impact between these cities and megacities based on population. Some of the observations and limitations in this study are as follows.

The initial understanding of this study was that urban land masses at the same latitude experience temporal shifts in seasons. It was observed that the highest LST values

occurred in the pairs of megacities and cities lying in the mid-latitudinal region. Seasonal variations not only depend on the location but also on topography, oceanic currents, and atmospheric wind disturbances in coastal regions. In some cases, the heat radiated from the built environment might affect oceanic currents and vice versa to create a much greater disturbance in the global context. Megacities in developing nations are facing considerable pressure in handling the cost of global warming, even though these nations are contributing less to the heat generated globally. It has been quantified that the megacities New York, Manila, Mumbai, and Karachi are generating more clusters of urban heat pockets than the similar latitudinal mid-sized cities (Figure 8). These mid-sized cities are very high in number compared to the existing megacities in the world. Soon they will be contributing to greater global heating issues. Tijuana, Tunis, Miami, and Dakar are in the process of becoming the hottest megacities in next decade due to their urban thermal performances, which are exceptionally similar to the existing megacities in the world. These upcoming megacities should follow the planning policy and urban design of Shanghai, the second most populated coastal city in the world; it maintains a maximum urban temperature below 35 °C, which is impressive.

**Author Contributions:** Conceptualization, D.H. and R.D.G.; methodology, D.H. and A.F.; software, D.H.; validation, D.H. and A.F.; formal analysis, D.H. and A.F.; investigation, D.H.; resources, R.D.G.; data curation, D.H.; writing—original draft preparation, D.H.; writing—review and editing, R.D.G.; visualization, D.H.; supervision, R.D.G.; project administration, A.F.; funding acquisition, A.F. All authors have read and agreed to the published version of the manuscript.

**Funding:** This research is partially funded by the Ministry of Education of the Government of India for first and second author, and for third author the research is partially supported by the Ministry of Science and Higher Education of the Russian Federation under the strategic academic leadership program "Priority 2030" (Agreement 075-15-2021-1333 dated 30 September 2021).

**Data Availability Statement:** The Landsat 8 data used in this study are openly available at https://earthexplorer.usgs.gov/ (accessed on 1 May 2022).

**Conflicts of Interest:** The authors declare no conflict of interest.

## Appendix A

Built/un-built maps with corresponding heat maps for both megacities and their respective mid-sized cities, as per the latitudinal reference, are shown for Ho Chi Minh City and Barranquilla, Manila and Dakar, and Karachi and Miami (Figure A1) as well as Shanghai and Tijuana, Tokyo and Tunis, and Vancouver and London (Figure A2).

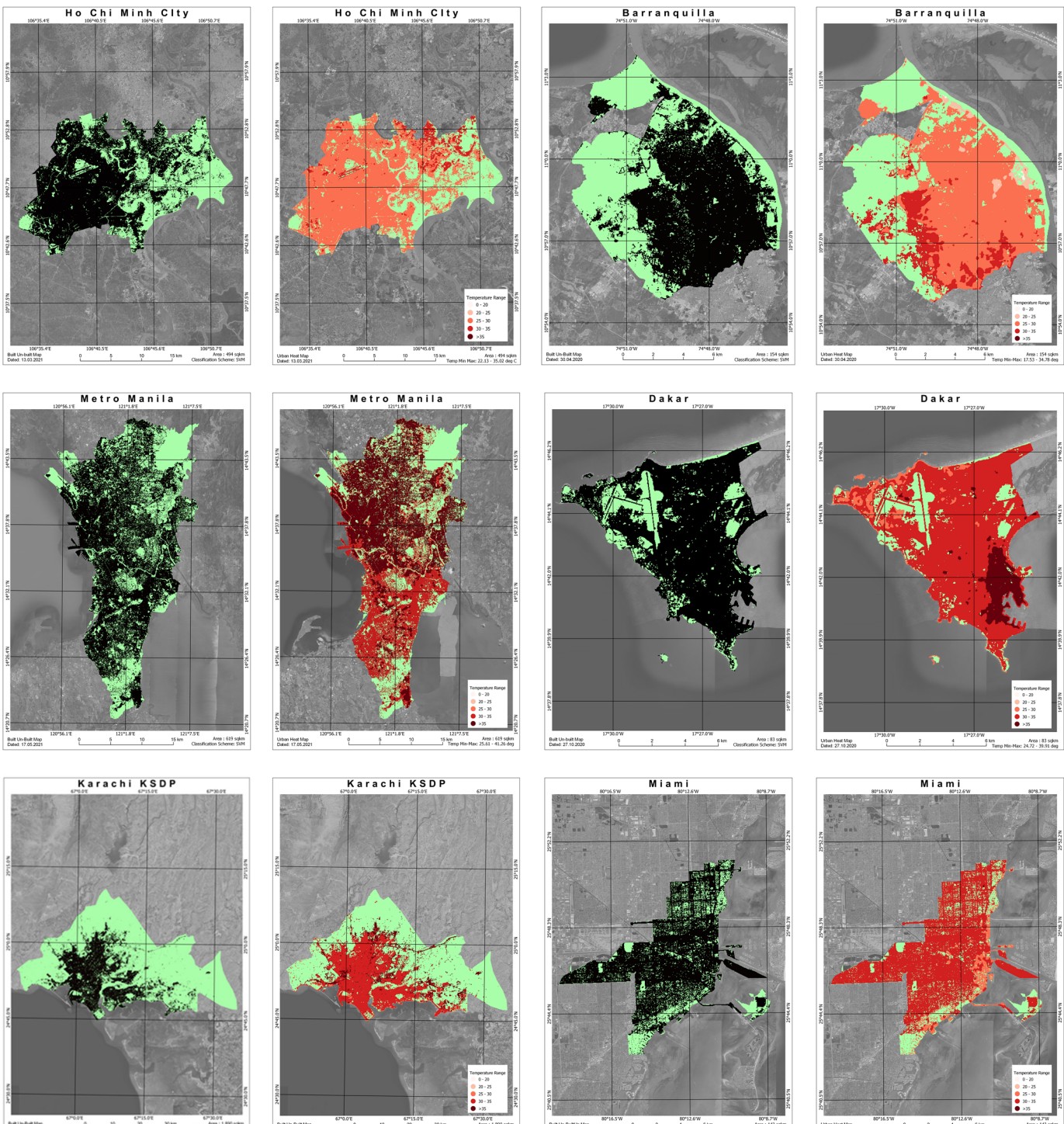

**Figure A1.** Built/unbuilt maps with corresponding heat maps for both megacities and their respective mid-sized cities, as per the latitudinal reference (**top**: Ho Chi Minh City and Barranquilla) (**middle**: Manila and Dakar) (**bottom**: Karachi and Miami).

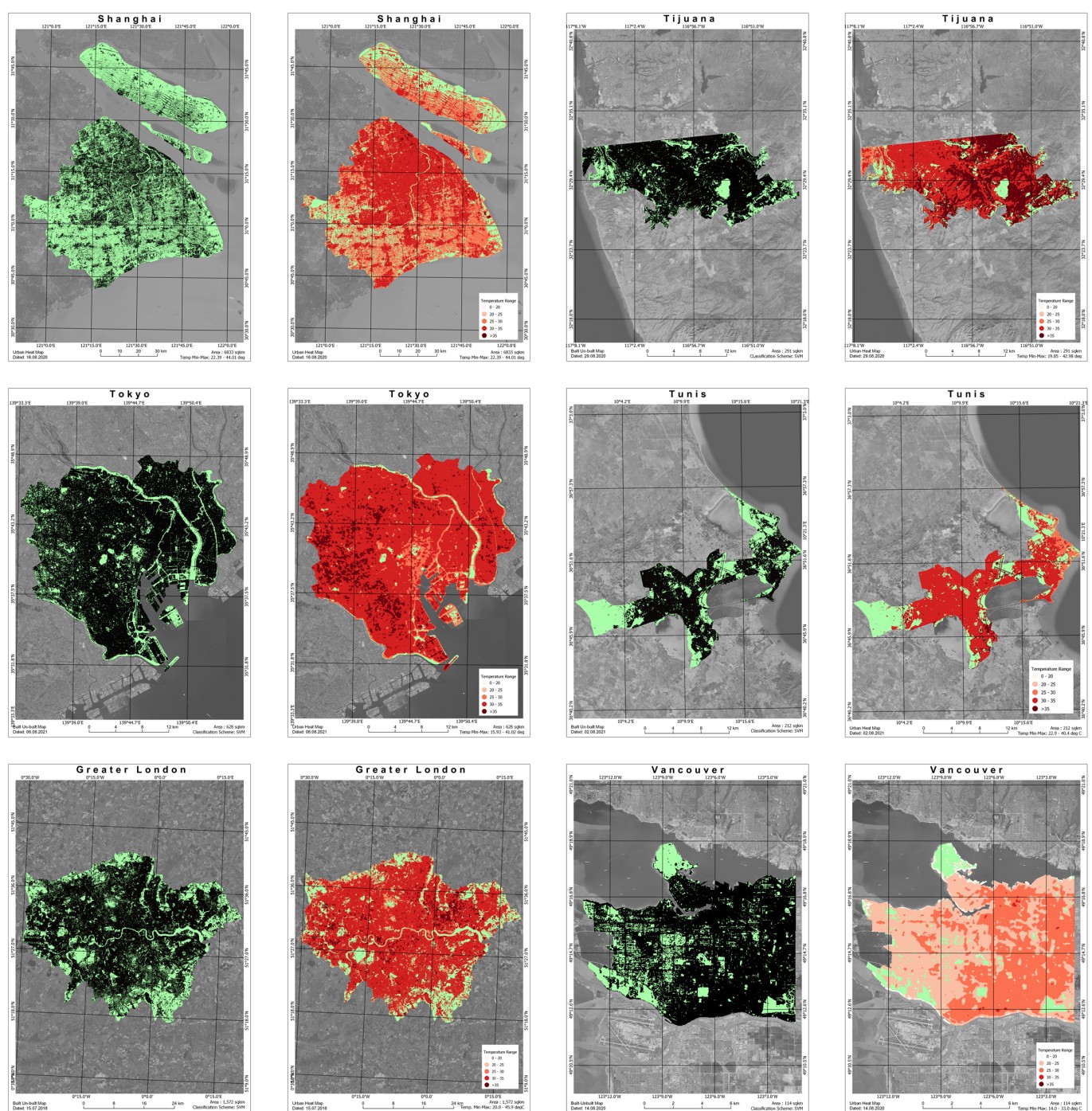

**Figure A2.** Built/unbuilt maps with corresponding heat maps for both megacities and their respective mid-sized cities, as per the latitudinal reference (**top**: Shanghai and Tijuana) (**middle**: Tokyo and Tunis) (**bottom**: Vancouver and London).

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
