# Peer review of "Latitudinal Trend Analysis of Land Surface Temperature to Identify Urban Heat Pockets in Global Coastal Megacities"

_remotesensing, doi:10.3390/rs15051355_

Round 1
Reviewer 1 Report
The first half of the abstract needs to be reworked to be more clear, understandable and readable. The link between sea level rise, storms and UHI is not clear at all.
"Given the significant uncertainties, a global rise in sea level by 2 metres until 2100 seems plausible [8, 9, 10]”: this is not true. Refs 8 and 9 are back from 1992 and 1993. This is by far too old in the field of Climate change. Ref 10 is awkward here. Please give proper scientific references from IPCC (AR6) or else.
Please add significant references for “local climate zone (LCZ)”. And explain this concept and why it is important here in this paragraph.
“Since the colonization”: what it is about? Please explain.
“from the unknowns”: please explain. UFO?
§ 109 – 116: please remove, not so interesting into a scientific article
“Mid-sized cities at similar latitudes are considered in this study for comparison”: why? What is the purpose of this comparison? How are the chosen? What are the criteria? Do the authors consider the proper climate of each city or only the location?
“Landsat-8 OLI and TM (Landsat-8 OLI TIR 30m) dataset are collected”: TM means Thematic Mapper and is on Landsat 7. There is confusion amongst the authors.
“from the USGS website”: which one exactly?
It seems that authors are using Landsat 8 Level 1 (or level 2 ?) collection 1 and calculating LST. Why not simply using Level 2 collection 2 with direct LST (nearly) ready to use?
“Afterwards, supervised classification was performed over all the datasets using SVM algorithm, which performed better than all other existing classification algorithms mainly in densely populated urban areas”: why? Purpose? References?
“For each image 100 – 300 ROI samples”: ROI has to be explained for all the readers that would not be familiar with it.
“The classes for the training samples were Built and Unbuilt”: what do the authors mean by “unbuilt”. Is a park unbuilt? Is it a green space area?
“barren land shows more LST values”: higher LST? Please improve English of the whole text
“they were reclassified in 5 classes based on the temperature defined by us as: >20, 20-25, 25-30, 30-35, >35”: why? Very arbitrary. And I guess this is <20.
“The most prevalent method for classifying satellite images is pixel-based classification, which uses quantitative approaches to discern patterns per pixel inside an image”: source? Other type of classification?
Typo: line 136
The visual quality of the equations have to be improved.
Fig. 3: cannot read the legend. Please improve
Fig. 3 to fig. 5: same kind of figures? Has to be placed into the appendix. Just keep a representative one.
Fig. 6: very confusing. A scatter plot would be better.
§ after fig. 6: looks like a catalog. Too descriptive. Please improve and remove the description sentences (“Karachi is the largest transport city in Pakistan, while Miami is largest tourist attraction spot and financial hub in Florida” – “Shanghai is the world’s largest city and Tijuana is a Mexican city adjacent to US border which makes it susceptible” (and susceptible to what), and so on).
“the places with same latitude will have same insolation as well as same thermal behavior”: I do not agree with the same thermal behavior. It depends on the local climatology (effect of the distance to the sea, the altitude, the local climate zones, the topography, the aspect, and so on), in addition to “local or regional climatic factors such as oceanic currents and wind pressure direction”.
I am not convinced about the pairing of the cities. For example, the climate of Tokyo is very different from the one in Tunis. Same for New York and Napoli or Vancouver and London). The authors should compare comparable cities with similar climate. They could start using the same Köppen-Geiger climate classification and then improve their choice using similarity statistics.
“As per the research, the outdoor comfortable temperature is 26.2°C, which is basically an air temperature; but the acceptable temperature with slight heat stress ranges between 26.2°C to 31.6°C [42]”: please, do more review about thermal comfort and give more than one subjective reference.
“26.2°C as day temperature at 0% humidity is an ideal condition”: 0% humidity? I do not agree at all. Please, do more serious review about thermal comfort.
The start of §4. is a broad discussion about thermal comfort that is not exact. Has to be totally reshaped. The end is very descriptive and looks like a catalog. I do not see much interesting discussion. The authors should have tried to cluster the different cities or something like that to get more efficient conclusions.
“The initial understanding of this study was that urban land masses at the same latitude may have temporal shift in seasons”: this is already well known in urban climatology and dynamic climatology.
It has been found out that the seasonal variations not only depend on the location but also on topography, oceanic currents, and atmospheric wind disturbance in coastal regions”: how? Where? Not in this article.
“The analogy of seasonal shift towardsequator, which suggests that temporally seasons are moving up from the conventional months has been found to be true”: I did not see that in that article
“In some cases, heat radiated from built environment is affecting oceanic currents and vice versa to create much greater disturbance in global context”: I did not see that in that article
“but it is well proved that megacities generate more severe urban heat pockets than mid-sized cities all over the globe”: this should not be part of a scientific article.
Author Response
Thank you for your comments and suggestions. We have made revisions according to it. Please find attached response to the comments. Revised portions are updated in the new version of the manuscript.

Reviewer 2 Report
The paper is interesting for the current problem of global warming especially in coastal cities where rising sea levels or the increase of storm formation could cause serious damages to the population.
I think the methodology is sufficiently explained and I have a few comments to improve the reading and focus the results.
Comments:
The Title ’Latitudinal trend analysis of Radiative Skin temperature’: the title is misleading I would use LST (land surface temperature) and not skin temperature
Abstract : Include some results
Line 57 : United Natin : I think it is Nation
Line 130 and 131 ; computation of classification over remotely sensed images using support vector machine : It is not clear classification of wath?
Table 2 : I did not understand the column severity (data and data1 )
Line 340 : According to the ideology : perhaps the author means methodology ?
Line 346-347 : Summer 2020 timeline was considered for LST extraction. In rare case, for better data analyses images were taken from summer of 2021/2019/2018 due to non- availability of cloud free data: could the authors explain better this point? I think that probably an analysis on the average of summer images (for each mega city) for at least 5 years would be more characterising
Something wrong in the figures: you call Figure 8 before Figure 6 and 7 that are in results paragraph. I suggest to change their position in Discussion (then change the numeration) or call them before in results. It is very difficult to read in this way
Figure 7 is not present in the text.
Figure 8: what is the x axis? Surface temperature please add it
Author Response

(The authors gave the same response as above.)

Round 2
Reviewer 1 Report
The authors hardly took my comments into account. The article has thus scarcely been improved.

Author Response
Honorable Reviewer, thank you for your comments and suggestions. We have made revisions according to it. Please find attached response to the comments. Revised portions are updated in the new version of the manuscript. Thank you for your time.
Sincere Regards
